# Bevacizumab-Based Chemotherapy Triggers Immunological Effects in Responding Multi-Treated Recurrent Ovarian Cancer Patients by Favoring the Recruitment of Effector T Cell Subsets

**DOI:** 10.3390/jcm8030380

**Published:** 2019-03-18

**Authors:** Chiara Napoletano, Ilary Ruscito, Filippo Bellati, Ilaria Grazia Zizzari, Hassan Rahimi, Maria Luisa Gasparri, Morena Antonilli, Pierluigi Benedetti Panici, Aurelia Rughetti, Marianna Nuti

**Affiliations:** 1Laboratory of Tumor Immunology and Cell Therapy, Department of Experimental Medicine, Sapienza University of Rome, Viale Regina Elena 324, 00161 Rome, Italy; ilary.ruscito@uniroma1.it (I.R.); ilaria.zizzari@uniroma1.it (I.G.Z.); hassan.rahimi@uniroma1.it (H.R.); aurelia.rughetti@uniroma1.it (A.R.); marianna.nuti@uniroma1.it (M.N.); 2Tumor Bank Ovarian Cancer Network (TOC), Department of Gynecology, European Competence Center for Ovarian Cancer, Campus Virchow Klinikum, Charité—Universitätsmedizin Berlin, Corporate Member of Freie Universität Berlin, Humboldt-Universität zu Berlin and Berlin Institute of Health, Augustenburger Platz 1, D-13353 Berlin, Germany; 3Department of Medical and Surgical Sciences and Translational Medicine, Sant’Andrea University Hospital, Sapienza University of Rome, Via di Grottarossa 1035, 00189 Rome, Italy; Filippo.bellati@uniroma1.it; 4Department of Maternal and Child and Urological Sciences, Policlinico Umberto I “Sapienza” University of Rome, Viale del Policlinico 155, 00161 Rome, Italy. M.antonilli@gmail.com (M.A.); pierluigi.benedettipanici@uniroma1.it (P.B.P.); Marialuisa.gasparri@uniroma1.it (M.L.G)

**Keywords:** bevacizumab, ovarian cancer, effector T cells, target therapy, immunotherapy, chemotherapy

## Abstract

Increasing evidence strongly suggests that bevacizumab compound impacts the immunological signature of cancer patients and normalizes tumor vasculature. This study aims to investigate the correlation between the clinical response to bevacizumab-based chemotherapy and the improvement of immune fitness of multi-treated ovarian cancer patients. Peripheral blood mononuclear cells (PBMCs) of 20 consecutive recurrent ovarian cancer patients retrospectively selected to have received bevacizumab or non-bevacizumab-based chemotherapy (Bev group and Ctrl group, respectively) were analyzed. CD4, CD8, and regulatory T cell (Treg) subsets were monitored at the beginning (T0) and after three and six cycles of treatment, together with IL10 production. A lower activated and resting Treg subset was found in the Bev group compared with the Ctrl group until the third therapy cycle, suggesting a reduced immunosuppressive signature. Indeed, clinically responding patients in the Bev group showed a high percentage of non-suppressive Treg and a significant lower IL10 production compared with non-responding patients in the Bev group after three cycles. Furthermore, clinically responding patients showed a discrete population of effector T cell at T0 independent of the therapeutic regimen. This subset was maintained throughout the therapy in only the Bev group. This study evidences that bevacizumab could affect the clinical response of cancer patients, reducing the percentage of Treg and sustaining the circulation of the effector T cells. Results also provide a first rationale regarding the positive immunologic synergism of combining bevacizumab with immunotherapy in multi-treated ovarian cancer patients.

## 1. Introduction

Ovarian cancer still accounts for the highest mortality rate among all gynecological malignancies, with 295,414 estimated new cases and 184,799 estimated new deaths in 2018 worldwide [1].

During the last 10 years, bevacizumab, a monoclonal antibody that binds to vascular endothelial growth factor (VEGF), has revolutionized the treatment approach in ovarian cancer, obtaining US Food and Drug Administration (FDA)/ European Medicines Agency (EMA) approval in all advanced disease settings (multi-treated/compassionate, platinum-resistant/platinum-sensitive recurrent, and primary International Federation of Gynecology and Obstetrics (FIGO) stage IIIB-IV ovarian cancer) (www.fda.gov; www.ema.europa.eu).

The biological mechanism underpinning the clinical efficacy of bevacizumab addition in ovarian cancer setting is still a matter of intense investigation. Increasing evidence supports the hypothesis that this biological compound modulates patients’ immune system, reducing immunosuppression and activating acquired immunity [2]. Several authors have shown the immune effects of bevacizumab in multiple cancer settings, such as decrease in patients’ regulatory T cells (Tregs) [3,4] and expansion of B and T cells [2].

Multi-treated recurrent ovarian cancer patients constitute an extremely fragile disease setting. Their immune system is weakened by multiple lines/types of treatment strategies, and the succession of therapeutic choices for them is currently discussed without a common consensus among oncologists.

In this study, we show that the clinical response to bevacizumab-based chemotherapy in this poor-prognostic disease correlates with improvement of patients’ immune fitness, thus providing new evidence that the benefit of such treatment can be ascribed also to its fine immune modulation.

## 2. Experimental Section

### 2.1. Patient Selection

This retrospective study received institutional review board (IRB) approval and was carried out following the rules of the Declaration of Helsinki of 1975. Patients included in this study were treated at the Gynecologic Oncology Unit of the Department of Gynecology, Obstetrics and Urology (Sapienza University of Rome, Italy) between 2008 and 2012. Since 2007, all gynecological cancer patients admitted in this department have been regularly subjected to donation of peripheral blood samples every three cycles of chemotherapy treatment for research purposes with their written informed consent. Patients’ peripheral blood mononuclear cells (PBMCs) were regularly collected and stored in liquid nitrogen at the Laboratory of Tumor Immunology and Cell Therapy Unit, Department of Experimental Medicine (Sapienza University of Rome, Italy—Ethical Committee approval, protocol n° 703/2008; date of approval 07/24/2008).

For this study, 20 consecutive recurrent ovarian cancer patients were retrospectively selected from “Sapienza” PBMC sample collection. All available multi-treated platinum-resistant ovarian cancer patients subjected to intraperitoneal (i.p.) bevacizumab-based chemotherapy as compassionate use were selected (Bev group; 10 patients), together with 10 patients (Ctrl group) that received non-bevacizumab-based chemotherapy.

Inclusion criteria were as follows: primary diagnosis of advanced epithelial serous ovarian cancer; having been subjected to at least three previous chemotherapy lines; diagnosis of tumor progression confirmed by CT scan; presence of malignant ascites; life expectancy of at least three months; and availability of at least three PBMC samples per patient collected during the course of bevacizumab-based versus non-bevacizumab-based chemotherapy. Furthermore, patients of the Bev group were matched with the Ctrl group patients for age, tumor grading, FIGO stage, type of primary treatment strategy (primary debulking surgery versus neoadjuvant chemotherapy followed by interval debulking surgery), tumor residual at first surgery, and type of recurrence at the time of blood sampling in order to minimize selection bias and avoid misinterpretations of results.

Ten patients were identified as the ones that had received i.p. bevacizumab 5 mg/kg every 21 days immediately after paracentesis for treatment of malignant ascites [5,6] plus intravenous (intravenous injection, i.v.) monochemotherapy (cisplatin) [7], while 10 other patients were identified as been treated with i.v. monochemotherapy alone (paclitaxel, topotecan, pegylated liposomal doxorubicin, cisplatin). Patients’ clinicopathological data were retrieved from clinical charts. Disease progression was defined basing on the response evaluation criteria in solid tumors (RECIST) [8].

### 2.2. PBMC Purification

PBMCs were isolated from 12 to 14 mL of peripheral blood by Ficoll–Hypaque gradient (1077 g/mL; Pharmacia LKB, Sweden), obtaining a yield between 10 × 10^6^ and 12 × 10^6^ cells for each drawing and cryopreserved until use. Samples were taken before therapy (T0) and after three (III) and six (VI) cycles of therapy.

### 2.3. Cell Phenotype

Cell phenotype staining was performed using several directly conjugated monoclonal antibodies (MoAbs). T cells were incubated with the anti-CD8-PE-Cy5.5 (RPA-T8 clone), anti-CD3-PE (UCHT1 clone), anti-CCR7-FITC (150503 clone), and anti-CD45RA-APC (HI100 clone) MoAbs, all from Becton Dickinson (Franklin Lakes, NJ, USA). Tregs were stained with the anti-CD25-PE (M-A251clone), anti-CD45RA-APC (HI100 clone), anti-CD4-FITC (RPA-T4 clone), and anti-FOXP3-Alexa 647 (259D/C7 clone) MoAbs, all from Becton Dickinson. Cells were incubated with the conjugated MoAbs targeting extracellular antigens for 30 min at room temperature (RT) as indicated by the manufacturer’s instruction. The staining of intracellular antigen FOXP3 was performed after the cells’ permeabilization with the Human FOXP3 Buffer Set (Beckton Dickinson, Franklin Lakes, NJ, USA). After washing, at least 1 × 10^4^ events were evaluated using a FACSCanto flow cytometer (Becton Dickinson, Franklin Lakes, NJ, USA) running FACSDiva data acquisition and analysis software (Becton Dickinson, Franklin Lakes, NJ, USA). The percentages of CD4 and CD8 T cells were calculated with respect to the entire CD3 T cell population, while the percentage of Treg was evaluated with respect to CD4 T cells.

### 2.4. Intracellular Cytokine Staining

T cells were stimulated with the anti-CD3 (OKT3 clone, 1 μg/mL) (eBioscence, San Diego, CA, USA) and anti-CD28 (CD28.2 clone, 5 µg/mL) (BioLegend, San Diego, CA, USA) MoAbs for 16 h at 37 °C in the presence of Brefeldin (Sigma-Aldrich, St. Louis, MO, USA) (10 µg/mL). The staining of IL-10 positive cells was carried out by fixing the cells with 2% paraformaldehyde (Sigma-Aldrich, Saint Louis, MO, USA). Cells were than washed with phosphate-buffered saline (PBS) without Ca^2+^ and Mg^2+^ + 0.5% saponin (Sigma-Aldrich, Saint Louis, MO, USA) + 10% fetal bovine serum (FBS) (Sigma-Aldrich, Saint Louis, MO, USA) and incubated for 30 min with anti-IL-10-PE (JES3-19F1 clone) (BioLegend, San Diego, CA, USA) MoAb. Cells were analyzed using a FACSCanto flow cytometer (Becton Dickinson, Franklin Lakes, NJ, USA) running FACSDiva data acquisition and analysis software (Becton Dickinson, Franklin Lakes, NJ, USA).

### 2.5. Statistical Analysis

Statistical analysis was performed using Graphpad Prism version 6 (Graphpad Software, Inc., San Diego, CA, USA).

Descriptive statistics (average and standard deviation) were used to describe different groups of continuous data. Student’s *t*-test was used to compare groups of continuous variables. Groups of categorical data were compared using the Fisher’s exact test. Significance is indicated when *p* ≤ 0.05.

## 3. Results

### 3.1. Patients’ Characteristics and Clinical Response

Twenty patients met all inclusion criteria and were included in the study. Patients’ characteristics are listed in Table 1. As a result of patient matching, no differences in terms of clinicopathological variables as well the Eastern Cooperative Oncology Group (ECOG) performance status could be identified between the Bev group and the Ctrl group. At the time of blood sampling for immunological analysis, 12/20 women (60%) presented intraperitoneal tumor progression, whereas the remaining 3/20 (15%) and 5/20 (25%) patients were diagnosed with intraperitoneal plus retroperitoneal disease worsening and widespread tumor dissemination, respectively.

From a clinical point of view, and as confirmed by serial Ca125 serum levels (Appendix A), 50% (10/20) of patients were judged responders to chemotherapy after six cycles of treatment and were equally distributed in each group of interventions (5/10 in the Bev group and 5/10 in the Ctrl group).

### 3.2. Bevacizumab-Treated Patients Showed a Different Immunological Signature Compared with the Control Group

To understand whether bevacizumab treatment impacts the immunological status of ovarian cancer patients, the modulation of circulating CD4 and CD8 T cells was firstly analyzed in the Bev group and the Ctrl group before (T0) and after III and VI cycles of treatments (Figure 1A). Both CD4 and CD8 T cells played a critical role in the activation of an effective antitumor immunity. CD8 lymphocytes exerted their cytotoxic activity by eliminating tumor cells, while CD4 T lymphocytes sustained and maintained a CD8 T cell response by cytokine production [9]. A deficiency in the activation of one of these two populations induced the development of a failed immunity against the tumor. Results obtained from the cancer patients showed that therapies did not modify the percentage of CD4 and CD8 lymphocytes in both groups at different time points. CD4 T cells were significantly higher in the Bev group at T0 and III compared with the Ctrl goup, although this difference disappeared at the end of VI cycles. No difference was observed in CD8 T cells between the two groups, although the ratio CD4/CD8 remained high (>1) up to VI cycles in both groups, suggesting a predominance of CD4 T cells during therapies.

CD8 and CD4 T cells were concurrently analyzed for the expression of CCR7 and CD45RA molecules, which identify four different lymphocyte subsets: effector (CCR7^−^CD45RA^+^), naïve (CCR7^+^CD45RA^+^), central memory (CCR7^+^CD45RA^−^), and effector memory (CCR7^−^CD45RA^−^) T cells. Analyzing these T cell subpopulations in the Bev and Ctrl group patients, no significant difference throughout the treatment in each patient group and between the two groups were found (data not shown).

Finally, the percentage of Tregs was also examined following the expression of CD4, CD25, and FOXP3 markers (Figure 1B). In cancers, Tregs represent one of the most important T cell populations as they are able to suppress the activation and/or expansion of antitumor CD4 and CD8 T cells through cell–cell contact or by cytokine release [10]. A high percentage of Tregs is associated with a poor prognosis in different types of solid tumors [11,12]. In our setting of patients, the results demonstrated that the Ctrl group showed a significant decrease in total Tregs from T0 to VI cycles (36% vs. 31%, *p* = 0.03), while no difference was found in the Bev group throughout the therapy. Total Tregs were further analyzed according to the combined expression of CD25, FOXP3, and CD45RA markers, which identifies three important Treg subpopulations [10]: resting Treg (CD25^+^CD45RA^+^FOXP3^+^: rTregs), activated Tregs (CD25^high^CD45RA^-^FOXP3^-^: aTregs), and cytokine-secreting Tregs with no suppressive activity (CD25^+^CD45RA^-^FOXP3^+^: nsTregs). aTreg have been described as terminally differentiated cells that rapidly die after exerting their suppressive activity, whereas rTreg proliferate and convert into aTreg both in vitro and in vivo [10]. The analysis revealed that bevacizumab-treated patients showed a lower percentage of aTregs and rTregs compared with the Ctrl group at T0. This difference persisted until III cycles of treatment in the rTreg subset and disappeared after VI cycles, suggesting that these patients exhibited a less suppressive immunological profile compared with the Ctrl group at the beginning and in particular after III cycles of therapies.

### 3.3. Bevacizumab-Treated Patients Showed a Discrete CD4 Effector T Cell Population throughout the Treatment

Patients belonging to the Bev group and the Ctrl group were then divided in clinically responders (R) and clinically nonresponders (N-R) to therapy according to RECIST (Appendix A). The modulation of CD4 and CD8 T cell was initially evaluated in R and N-R patients of both groups, followed by the analysis of the different T cell subsets (Figure 2). The results demonstrated that the CD8 T cells derived from bevacizumab-treated patients were not differently modulated in R and N-R patients, while the CD4 T cells appeared to be significantly higher in the N-R group at the beginning and after VI cycles of treatment. Conversely, in the Ctrl group, the CD8 T cells seemed to be significantly higher after VI cycles in the R patients compared with the N-R ones, while no significant difference was observed in the CD4 T cell population.

Lymphocytes were also analyzed according to the expression of CCR7 and CD45RA molecules (Figure 3). The results demonstrated that in the Bev group and the Ctrl group, the percentage of CD4 effector T lymphocytes in R patients was higher compared with N-R patients at T0. This difference persisted until the end of the therapies for bevacizumab-treated patients, while it had already disappeared after III cycles of therapy in the control group. This suggests that Bev treatment, by favoring the normalization of the tumor vasculature [13], improves and sustains the circulation of effector T cells.

The other CD4 T cell subsets and the CD8 T cell populations were not significantly modified by treatments (data not shown).

### 3.4. Tregs Were Modulated in Bevacizumab-Treated Patients during Therapies

To assess whether the treatment schedule and/or the clinical response could be associated with the modulation of the Treg subsets, the percentage of circulating Tregs after III or VI treatment cycles were compared with the baseline value at T0, and the analysis was expressed as fold increase (%TregIII/%TregT0 or %TregVI/%TregT0) (Figure 4). After III cycles of treatment, the level of the entire Treg population was significantly higher in R patients compared with N-R patients in the Bev group. This increase was ascribed to the nsTreg subset being significantly higher in R compared with N-R patients. These differences between R and N-R patients disappeared after VI cycles of bevacizumab treatment. In contrast, the control group did not show any difference in the percentage of Tregs between R and N-R patients during therapies, and no difference between the Bev and Ctrl patients was observed.

### 3.5. Bevacizumab-Treated N-R Patients Had Higher Level of IL10^+^ T Cells Compared to R Patients

Because IL10, such as TGFβ, is one of the most important cytokines released by Tregs [14] that is able to downregulate Th1 cytokine production and block NF-κB activity [15], T cell derived from patients in the Bev group were analyzed for their capacity to produce IL10 as intracellular staining (Figure 5). These patients exhibited a significant increase in IL10^+^ cells from T0 up to VI cycles of therapy. Analyzing the data as fold increase of the percentage of IL10^+^ cells after III and VI cycles of therapy compared with T0 (%IL10_III_/%IL10_T0_ or %IL10_VI_/%IL10_T0_) between R and N-R patients, significant high levels of IL10 were found in N-R patients after III cycles, suggesting an enhancement of the immunosuppression during the bevacizumab treatment in this group. This increase disappeared after VI cycles (*p* = 0.08) of therapy.

## 4. Discussion

Multi-treated progressive ovarian cancer still remains the most challenging disease setting for gynecologic oncologists, and so far, no global consensus has been met about how/how long patients should be continued to be treated [16]. Indeed, all these patients progressively develop pharmacoresistance for the majority of conventional chemotherapy drugs, and the only option left (if they are not eligible for phase I clinical trials) is to retreat them with previously adopted compounds [17]. Among biological agents that have obtained FDA/EMA approval in compassionate ovarian cancer setting, bevacizumab has been the first to enter into clinical practice after showing its ability to improve patients’ quality of life and also reduce paracentesis frequency for women suffering malignant ascites [5].

The effects of bevacizumab on patients’ immune system are still not completely elucidated, although a strong rationale about the interplay between its ligand (VEGF) and the host’s immune response suppression has already been shown [18]. In particular, three different mechanisms related to VEGF-mediated immunosuppression have been assessed so far: inhibition of dendritic cell maturation [19,20,21], reduction of T cell tumor infiltration [22], and promotion of inhibitory cells in the tumor microenvironment [23].

In this scenario, the present study adds new evidence to the body of knowledge concerning the immune effects of bevacizumab in advanced cancer patients by showing that (1) ovarian cancer patients not treated with bevacizumab-based chemotherapy seem to have a more immunosuppressive profile with the presence of a rTreg population that persists until the end of III cycles of therapy; (2) patients that clinically respond to bevacizumab treatment show a discrete population of effector T cells at the beginning of therapy that is maintained throughout the treatment; (3) Tregs are mainly represented by non-suppressive regulatory T cells in clinically responding bevacizumab patients compared with nonresponding patients and are also stably maintained in this ratio (nsT reg > sup T reg) throughout the treatment; (4) after three cycles of treatment, nonresponding bevacizumab patients produce more immunosuppressive IL-10 cytokine compared with responding patients.

It should be pointed out that these results were obtained by comparing two groups of patients that were matched for all clinical characteristics. Particularly important is to notice that the performance status was similar among the two groups; this variable has been significantly associated with the immunological effects and response to several therapies [24].

Other authors have observed an impact of bevacizumab-based regimens on the immunosuppressive status of cancer patients in different cancer settings. In particular, it was recently reported that, for glioblastoma patients subjected to radiation plus temozolomide (TMZ) and bevacizumab, the absolute number of peripheral Tregs significantly decreased following treatment [25]. Furthermore, the addition of bevacizumab to standard radiation and TMZ appeared to decrease the number of circulating Tregs compared with radiation plus TMZ alone. On the contrary, they also noticed a significant decrease in the absolute number of cytotoxic CD8 (CD107a^+^), effector memory CD8, and naïve CD4 T cells in the group of bevacizumab-treated patients.

Our results follow and confirm the original observation by our group [6] in which a significant reduction of Tregs and an increase in the proportion and function of effector CD8 T cells were found in an end-of-life ovarian cancer patient treated with low-dose intraperitoneal bevacizumab for malignant ascites.

We also showed that responding bevacizumab-treated patients reported a higher percentage of circulating CD4 effector T cells compared with nonresponding bevacizumab patients, confirming what has already been observed in metastatic colorectal cancer [2]. This data has key implications in the current panorama of oncological clinical approach. Indeed, it is reasonable to suggest that the circulating effector T cells recruited and sustained by bevacizumab treatment, thanks to its ability to restore tumor microvascular normalization [13], could be expanded by the administration of checkpoint inhibitor agents, thus giving a strong biological rationale for the combination of immunotherapy with bevacizumab antiangiogenetic therapy. In support of this consideration, tumor tissue derived from metastatic renal cell carcinoma (mRCC) patients treated with anti-PD-L1 atezolizumab plus bevacizumab was recently found to show an increase in intratumoral CD8 T cells as well as an increase in intratumoral MHC-I, Th1 and T-effector markers, and chemokines. Trafficking lymphocytes also increased in tumors following bevacizumab and combination treatment [26].

We finally observed that bevacizumab responding patients showed significant lower circulating immunosuppressive IL-10 cytokine levels compared with non-responding patients, thus confirming the effect of bevacizumab in reducing patients’ immune suppression. A similar finding was recently obtained in breast cancer neoadjuvant setting [27]. Patients treated with bevacizumab-included neoadjuvant chemotherapy showed a global decrease in circulating cytokines levels, such as VEGF-A, IL-12, IP-10, and IL-10. In addition, the decrease in IL-10 serum levels was confirmed to be even greater in response to bevacizumab treatment in metastatic colorectal cancer setting [28].

To our knowledge, this is the first study analyzing the immunological effects of bevacizumab-based treatment in women with advanced ovarian cancer in relation to their clinical response.

However, several significant limitations are present in this study. The first is the restricted number of patients that were retrospectively and not randomized selected. Moreover, there were no data regarding the immune fitness of the patients at diagnosis before the beginning of all therapies. Finally, although no differences in clinicopathological variables were identified, the two groups analyzed showed several differences in their immunological signature at T0. These differences could be ascribed to the several chemotherapy treatments (e.g., taxol, bemcitabine, pegilated liposomal, doxorubicin etc.) that differently impact the immunological system [29], together with the capacity of the immunological signature of each patient to differently respond to the same environmental factor, such as chemotherapy and surgery. In this setting, a discrete population of effector CD4 T cell is present in any case in both populations of patients independently from previous treatment. This cell subset is the one that appears to be affected by the bevacizumab regimen, and this could impact clinical outcome.

Another important point is the occurrence of leukopenia in multi-treated patients, which represents an important prognostic factor in patients with advance malignances [30]. In our setting, patients were affected by mild/moderate leukopenia that was non-clinically significant.

The strengths of this study can be summarized as follows: (1) the population involved was homogeneous for histology and clinical stage; (2) the patients belonging to the two groups (bevacizumab-treated and control) were matched for age, tumor grading, FIGO stage, type of primary treatment strategy (primary debulking surgery versus neoadjuvant chemotherapy followed by interval debulking surgery), tumor residual at first surgery, and type of recurrence at the time of blood sampling, thus minimizing selection bias; (3) blood sampling was carried out for all patients at the same times of treatment, i.e., at T0 and after three and six cycles of therapy.

## 5. Conclusions

In conclusion, this study sheds a light on the strong need to routinely include immunomonitoring in oncological clinical protocols of patients at follow-up during the course of antiangiogenetic therapy administrations, with the final aim being to identify early the subset of patients who can mostly benefit from its adoption. Furthermore, the study provides a first rationale regarding the positive immunologic impact of combining bevacizumab with checkpoint inhibitors. Confirmatory studies carried out on larger cancer patient populations are warranted.

## Figures and Tables

**Figure 1 jcm-08-00380-f001:**
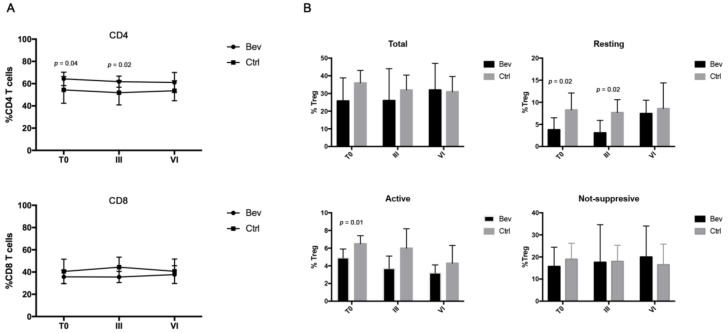
Evaluation of CD4 and CD8 T cell in the bevacizumab (Bev) group and the control (Ctrl) group by cytofluorimetry. (**A**) Analysis of the percentage of CD4 and CD8 T cells derived from patients belonging to the Bev group and the Ctrl group before (T0) and after III and VI cycles of therapies. CD8 T cells were identified by gating the CD3^+^CD8^+^ cells, while the CD4 T cells were identified as CD3^+^CD8^−^. (**B**) Histograms represent the percentage of the different regulatory T cell subset (total, active, resting, and nonsuppressive) calculated on CD4^+^CD25^+^cells. Bev group and Ctrl group are represented with black and grey histograms, respectively.

**Figure 2 jcm-08-00380-f002:**
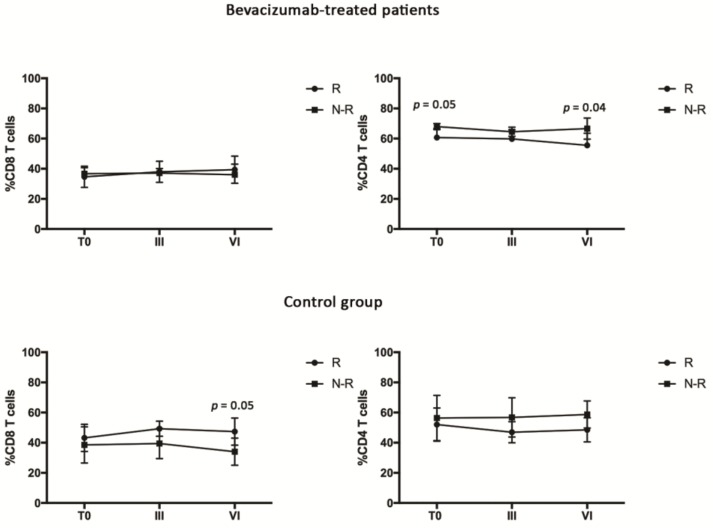
Evaluation of CD4 and CD8 T cells in responding (R) and nonresponding (N-R) patients of the bevacizumab-treated group and the control group by cytofluorimetry. CD8 T cells were identified by gating the CD3^+^CD8^+^ cells, while the CD4 T cells were identified as CD3^+^CD8^−^.

**Figure 3 jcm-08-00380-f003:**
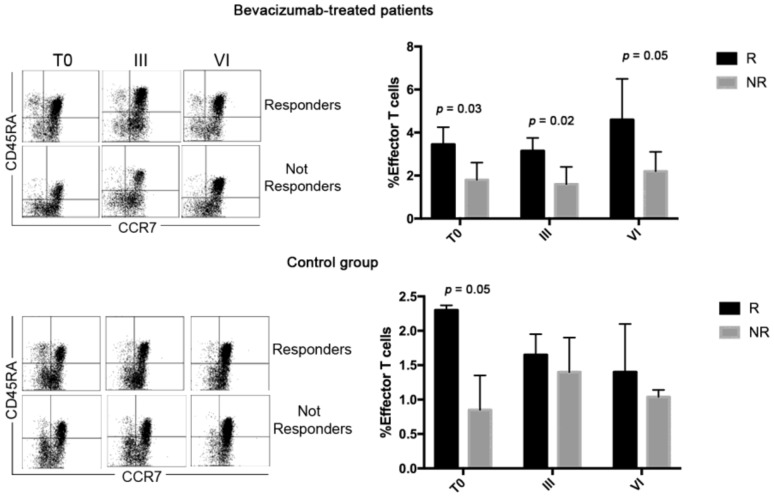
Analysis of CD4 and CD8 T cell subsets were carried out using the anti-CD3, anti-CD8, anti-CCR7, and anti-CD45RA MoAbs. CD8 T cells were identified by gating the CD3^+^CD8^+^ cells, while the CD4 T cells were identified as CD3^+^CD8^−^. Dot plots show the expression of CD45RA and CCR7 molecules that identify different T cell subsets (T effector: CD45RA^+^CCR7^−^; T central memory: CD45RA^−^CCR7^+^, T naive: CD45RA^+^CCR7^+^, and T effector memory: CD45RA^−^CCR7^−^) at T0 and after III and VI cycles of therapy. Histograms represent the median values of the percentage of effector T cells (CD45RA^+^CCR7^−^) of 10 patients (five patients of R and N-R of both Bev group and Ctrl group) ± standard deviation. Black and grey columns correspond to responding and nonresponding patients, respectively.

**Figure 4 jcm-08-00380-f004:**
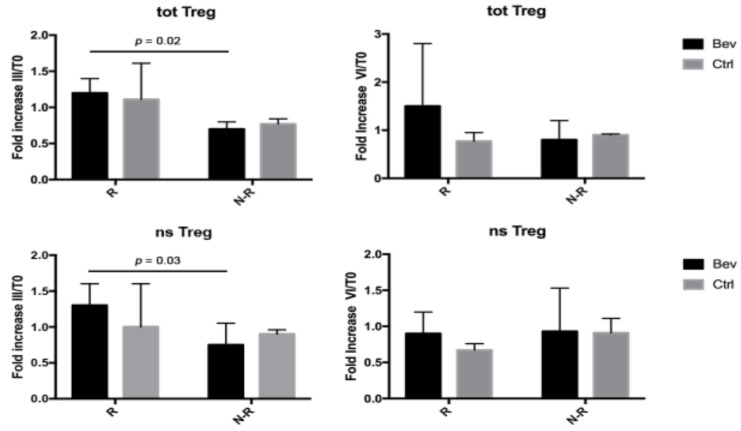
Total and non-suppressive Treg (totTreg and nsTreg, respectively) evaluated as fold increase after III or VI cycles of therapy compared with T0 (%TregIII or %TregVI/%TregT0). Black and grey columns correspond to Bev group and Ctrl group, respectively.

**Figure 5 jcm-08-00380-f005:**
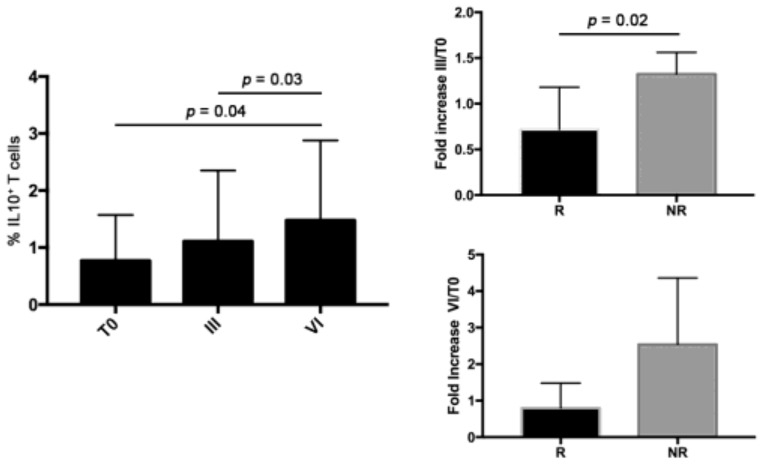
Evaluation of IL10^+^cells at T0 and after III and VI cycles of treatment in bevacizumab patients and fold increase of the percentage of IL10^+^cells after III and VI cycles of therapy compared with T0 (%IL10 III or %IL10 VI/%IL10 T0) in R and N-R patients belonging to the Bev group.

**Table 1 jcm-08-00380-t001:** Patients’ characteristics.

	Bevacizumab-Treated Patients	Control Group	*p*-Value
Patient n°	10	10	
Age (median, range)	54 years (42y–67y)	48.5 years (45y–71y)	0.845
ECOG Performance Status			
1	1/10 (10%)	2/10 (20%)	
2	7/10 (70%)	5/10 (50%)	
3	2/10 (20%)	3/10 (30%)	
Tumor Grading at primary diagnosis			0.628
I	0	0	
II	4/10 (40%)	2/10 (20%)	
III	6/10 (60%)	8/10 (80%)	
FIGO stage at primary diagnosis			1
IIIC	8/10 (80%)	7/10 (70%)	
IV	2/10 (20%)	3/10 (30%)	
PDS NACT	5/10 (50%) 5/10 (50%)	6/10 (60%) 4/10 (40%)	1
RT at first surgery (cm)			1
=0	9/10 (90%)	8/10 (80%)	
>0	1/10 (10%)	2/10 (20%)	
Type of recurrence at the time of blood sampling			0.061
Intraperitoneal only	7/10 (70%)	5/10 (50%)	
intraperitoneal + retroperitoneal	1/10 (10%)	2/10 (20%)	
widespread	2/10 (20%)	3/10 (30%)	

NACT: Neoadjuvant chemotherapy; PDS: Primary Debulking Surgery; RT: Residual Tumor.

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
