# Peer review of "Bevacizumab-Based Chemotherapy Triggers Immunological Effects in Responding Multi-Treated Recurrent Ovarian Cancer Patients by Favoring the Recruitment of Effector T Cell Subsets"

_jcm, 2019, doi:10.3390/jcm8030380_

Reviewer 1 Report

This manuscript is described of chemotherapy triggers immunological effects in  recurrent ovarian cancer patients, well written and interesting.

I think this manuscript is acceptable.
But I have minor concern before publication.

 Experimental Section

・L96  Is the dose of bevacizumab 5 mg / kg every 21 days?

・L82  Please describe how did you selected 20 patients. Did you select it randomly? Why did you selected only 20 patients? I think that it is better to analyze again for all patients from 2008 to 2012.

・Author should change the Ref (8) to follow as:

Eisenhauer EA,et al. New response evaluation criteria in solid tumours: revised RECIST guideline (version 1.1). Eur J Cancer. 2009 Jan;45(2):228-47.

 Results

・I think that it is better to add Performance status (PS) to the Table1. Data of immunological effects may be different depending on PS difference. Therefore it is better to state that there is no difference in PS between both groups.

 Author Response

Point 1: L96 Is the dose of bevacizumab 5mg/kg every 21 days?

 Response 1: yes, in the section "Materials and Methods" the Authors reported:

 "Ten patients were identified as the ones that had received intraperitoneal (i.p.) bevacizumab 5g/kg every 21 days, immediately after paracentesis, for malignant ascites treatment [5,6], plus intravenous (i.v.) mono-chemotherapy (cisplatin) [7]".

The amount of 5g/kg was erroneously written, this typing error has been corrected into 5mg/kg.

 Point 2: L82, Please describe how did you selected 20 patients. Did you select it randomly? Why did you selected only 20 patients? I think that it is better to analyse again for all patients from 2008 to 2012.

 Response 2: All available multi-treated platinum resistant ovarian cancer patients subjected to bevacizumab-based chemotherapy from 2008 to 2012 were selected from Sapienza PBMC collection (10 patients). These patients were treated intraperitoneally as compassionate use and were the only one’s available between 2008-20012. An equal number of multi-treated platinum resistant ovarian cancer patients subjected to other conventional regimens were selected as control (10 patients). A sentence has been added in the Material and Method section (Line 82-85).

 Point 3: Author should change the Ref (8) to follow as: Eisenhauer EA,et al. New response evaluation criteria in solid tumours: revised RECIST guideline (version 1.1). Eur J Cancer. 2009 Jan;45(2):228-47.

 Response 3: Reference 8 has been changed

 Point 4: I think that it is better to add Performance status (PS) to the Table1. Data of immunological effects may be different depending on PS difference. Therefore, it is better to state that there is no difference in PS between both groups.

 Response 4: ECOG PS has been added in table 1. P value reported in table 1 confirms there is no difference between both groups in terms of PS.

A comment regarding the PS has been added in the discussion (Line 295-298)

Reviewer 2 Report

In the present study, the authors aimed to investigate the correlation between the clinical response to bevacizumab-based-chemotherapy and the improvement of immune fitness of multi-treated-ovarian cancer patients.

A significant selection bias exists in this retrospective study since patients were not randomized into two groups (Bev-group and Ctrl-group). Although no differences in clinical-pathological variables could be identified between the two groups, Bev-group showed some different immunological signatures at T0 (before treatment), such as CD4 T cells, aTregs and rTregs, compared to Ctrl-group. Thus it is very difficult to identify what immunological signatures were actually affected by bevacizumab-based chemotherapy.

Absolute numbers of peripheral lymphocytes at T0, III, and, VI should be indicated, because leukocytopenia is often observed in patients with heavily pretreated ovarian cancer and, in addition, pretreatment lymphopenia is a prognostic marker in patients with advanced malignancies.[Cancer Res 2009;69:5383] 

The authors must indicate what represents “%Tregs.” Does “%Tregs” represent the percentage of Tregs among the total CD4+ T cells? Similarly, %CD4 T cells and %CD8 T must be explained.

Author Response

Point 1: A significant selection bias exists in this retrospective study since patients were not randomized into two groups (Bev-group and Ctrl-group). Although no differences in clinical-pathological variables could be identified between the two groups, Bev-group showed some different immunological signatures at T0 (before treatment), such as CD4 T cells, aTregs and rTregs, compared to Ctrl-group. Thus it is very difficult to identify what immunological signatures were actually affected by bevacizumab-based chemotherapy.

 Response 1: Data reported in the paper showed that at T0 Bev-group had a higher percentage of CD4 T cells compared to Ctrl-group and a lower percentage of resting and active Treg. Although these patients showed no clinical-pathological variables, we should keep in mind that the patients belonging to both groups were subjected to different number and type of chemotherapy regiments (e.g. Taxol, gemcitabine, pegylated liposomal, doxorubicin etc) that differently impact the immunological system as describe by Zitvogel et al (Nat Rev Immunol, 2008). In addition, it is important to consider that the immunological signature of each patient could differently respond to the same environmental factor, such as chemotherapy and surgery. These two important factors could explain the differences observed between the two groups.

This comment has been added in the Discussion (line 332-338)

 Point 2: Absolute numbers of peripheral lymphocytes at T0, III, and, VI should be indicated, because leukocytopenia is often observed in patients with heavily pretreated ovarian cancer and, in addition, pretreatment lymphopenia is a prognostic marker in patients with advanced malignancies.[Cancer Res 2009;69:5383].

 Response 2: We agree that leukopenia represent an important prognostic factor. We only carried out palliative chemotherapy in patients with mild/moderate non-clinically significant leukopenia. However, the objective of the presence study was to analyse the lymphocyte population and not general white blood cell count and data is unavailable due to the fact that blood exams were carried out in external laboratories. We have added a paragraph in the discussion (line 339-341).

 Point 3: The authors must indicate what represents “%Tregs.” Does “%Tregs” represent the percentage of Tregs among the total CD4+ T cells? Similarly, %CD4 T cells and %CD8 T must be explained.

 Response 3: The information regarding the gating strategies has been added in the Material and Method section (Line 121-123) as described below:

“The percentages of CD4 and CD8 T cells were calculated respect to entire CD3 T cell population, while the percentage of Treg was evaluated respect to CD4 T cells”.

Round  2

Reviewer 2 Report

  If lines 332-341 are crossed out in the manuscript, the authors should mention the limitations of this study. 

Author Response

Point 1: the authors should mention the limitations of this study. 

Response 1: To better specify the limitation of the study, we have added the below paragraph to the discussion (line 332-341)

However, several significant limitations are present in this study. First of all, the restricted number of patients that were retrospectively and not randomized selected. Moreover, there were no data regarding the immune fitness of the patients at diagnosis before the beginning of all therapies. Finally, although no differences in clinical-pathological variables were identified, the two groups analyzed showed several differences in their immunological signature at T0. These differences could be ascribed to the several chemotherapy treatments (e.g. taxol, bemcitabine, pegilated liposomal, doxorubicin etc) that differently impact the immunological system [29], together with the capacity of the immunological signature of each patient to differently respond to the same environmental factor, such as chemotherapy and surgery. In this setting, a discrete population of effector CD4 T cell is present in any case in both populations of patients independently from previous treatment. This cell subset is the one that appears to be affected by bevacizumab regimen and that could impact clinical outcome.
